# A Phytogeographical Classification and Survey of the Indigenous Browse Flora of South Africa, Lesotho, and Eswatini

Marike Trytsman [1,*], Francuois L. Müller [1], M. Igshaan Samuels [1,2], Clement F. Cupido [1] and Abraham E. van Wyk [3,4]

1    Agricultural Research Council—Animal Production: Rangeland and Forage Sciences, Pretoria 0001, South Africa; mullerf@arc.agric.za (F.L.M.); samuelsi@arc.agric.za (M.I.S.); cupidoc@arc.agric.za (C.F.C.)

2    Department of Biodiversity and Conservation Biology, University of the Western Cape, Cape Town 7535, South Africa

3    Department of Plant and Soil Sciences, University of Pretoria, Pretoria 0028, South Africa; braamvanwyk@gmail.com

4    National Herbarium, South African National Biodiversity Institute, Pretoria 0001, South Africa

\*    Correspondence: mtrytsman@arc.agric.za

**Abstract:** Rangelands in South Africa, Lesotho, and Eswatini contain a rich diversity of valuable fodder trees and shrubs. This research is the first attempt to document the regional diversity and distribution of these browse resources. Scientific publications, textbooks, databases, and published reports were accessed to compile a database of plant species that were recorded as utilised by ruminants and non-ruminants. Relevant forage attributes, such as functional traits as well as utilisation traits, were added to each species record. Thereafter, distribution records were extracted from the South African National Biodiversity Institute's Botanical Database of South Africa and analysed with numerical techniques to establish phytogeographical patterns. A total of 613 plant species from 76 families have been recorded, which formed seven distinct phytochoria, termed the Central Arid, Eastern Subtropical, Highland Temperate, Moist Temperate, Northern Subtropical, Southern Temperate, and Western Arid browse-choria. Key families and species, as well as functional and utilisation traits, are discussed, focusing on key species present in the browse-choria. This browse database, together with the earlier compiled Leguminosae and Poaceae databases, will be used to prioritise indigenous southern African plant species/infraspecific taxa to be collected for the conservation of genetic resources and future evaluations for potential development as forage crops.

**Keywords:** conservation; diversity; forage; game; livestock; phytochoria; rangelands; wildlife

## 1. Introduction

Rangelands cover >70% of South Africa's land surface [1], which is much higher than the total cover of about 54% for the globe [2]. In South Africa, rangelands are primarily utilised for extensive livestock production, game farming, and/or nature conservation activities [3]. These rangelands contain a rich diversity of tree and shrub species that are utilised as fodder by both livestock and indigenous wildlife. Collectively, these fodder plants (mostly perennials) are referred to as browse and include trees, shrubs, and dwarf shrubs [4]. Browsers in South Africa include animals such as domestic goats (*Capra aegagrus hircus*), giraffe (*Giraffa camelopardalis*), and kudu (*Tragelaphus strepsiceros*), whereas mixed feeders are, e.g., gemsbok (*Oryx gazella*), domestic sheep (*Ovis aries*), and African elephant (*Loxodonta africana*) [5–7]. On occasion, especially when forage is limited, cattle (*Bos taurus*, broadly defined) also consume browse [8], but its value may also include benefits such as endoparasite control [9].

From an agricultural perspective, these browse resources are important components of extensive livestock production and game farming activities, where browse is directly

utilised by both mixed feeders (30–70% grass or browse, which could change seasonally) and browsers (>80% browse and wild fruit) [6]. The extent of utilisation, however, depends on the physical and chemical properties of the species and the ability of the animals using these resources to successfully access and digest the fodder. This is because browse has both physical (e.g., spinescence) and chemical (e.g., secondary compounds) antiherbivore defenses, as well as significant variations in their chemical composition that affect their palatability [10,11]. Seasonal variations in acceptability also play a role in the utilisation of browse [12], but the flexibility of particularly bovids in their food preference, driven by changes in nutritional demands and food availability, is emphasised by Radloff et al. [13]. Owen-Smith and Cooper [14] concluded that domesticated goats are less selective between woody browse species compared to certain indigenous wildlife species and that some browse is utilised year-round, while others are mostly rejected, except during dry periods when it is the only forage available. These authors further found that, in a deciduous savanna, browse such as *Diospyros lycioides* and *Grewia flavescens* were preferentially utilised by browsers such as kudu during the wet season, while the same browsers preferentially used *Searsia leptodictya* and *Strychnos pungens* during the dry season.

The importance of browse resources for livestock and game farming in South Africa requires us to look more closely at these natural resources, not only from an indigenous fodder resource or fodder production perspective but also from a biodiversity perspective. Some indigenous browse species, however, are problematic and proliferate in rangelands through a process called bush encroachment [15,16]. It is estimated that 7.3 million ha of South Africa's land area (i.e., 6% of the country) is affected by bush encroachment [17]. In an attempt to effectively manage these browse resources, a better understanding of the phytogeographical patterns of these browse resources is needed. Browse forage resources, however, are spatially (horizontally and vertically) more complex compared to grass forage resources due to adjoining assemblages of browse species and the fact that several species can occupy a large range of bioclimatic and edaphic conditions [11].

Walker [18] distinguished seven broadly demarcated woody vegetation types in southern and south-tropical Africa, namely *Acacia* Savanna (African *Acacia* now classified as either *Senegalia* or *Vachellia*), Arid Shrub and Tree Savanna, *Baikiaea* Woodlands (Zambia), *Colophospermum mopane* Savanna, Karoo, Mixed Tree and Shrub Savanna, and, lastly, Miombo Woodlands (south-tropical Africa). The Savanna is broadly divided by Huntley [19] into a moist/dystrophic (broad-leaved) and an arid/eutrophic (fine-leaved) savanna. The former, according to Rutherford et al. [20], forms part of the higher-lying areas of the Central Bushveld Bioregion, and the latter of the lower-lying parts of the Central Bushveld, Lowveld, Sub-Escarpment Savanna, Mopane, Eastern Kalahari Bushveld, and Kalahari Duneveld bioregions.

Apart from the contributions cited above, very little further work has been performed on the phytographical classification of southern African woody vegetation types at the macroregional scale (barring many phytosociological studies at the microregional scale), and none has focused specifically on species that are utilised as browse by livestock and indigenous game. Note that references to "game" in the present contribution primarily pertain to larger mammals, many of which are hunted for commercial purposes. Therefore, this study aimed to describe (1) the phytogeographical patterns, whereby browse is divided into phytochoria, defined as a relatively large area with mostly homogeneous plant species (floristic) composition [21], (2) the functional attributes (height and morphology), and (3) the utilisation (plant parts and browser type) of browse indigenous to South Africa, Lesotho, and Eswatini, a combined region henceforth referred to as southern Africa.

In this study, browse is defined as a woody shrublet, dwarf shrub, shrub, or tree known to be utilised as food by livestock and game, both by ruminants and/or non-ruminants. The term browse-choria describes the phytochoria based on assemblages of distinctly woody species utilised by herbivores, similar to leguminochoria (assemblages of Leguminosae) and grasschoria (assemblages of Poaceae) referred to in previous studies [22–24]. The phytogeographical patterns and significance of the browse-choria (from this study), as well

as the previously recognised leguminochoria and grasschoria, will assist in formulating a collection, conservation, screening, and characterisation strategy for indigenous genetic resources with pasture and or soil conservation potential for the South African National Forage Genebank.

## 2. Materials and Methods

An inventory of indigenous plant species recorded as being browsed by livestock and game in southern Africa was compiled by accessing 24 scientific articles, five textbooks, two databases, and two published reports. Keywords used in searches included: shrubs, trees, browse, browser, game, livestock, wildlife, nutritional quality, utilisation, and South Africa. No publications were retrieved for Eswatini and Lesotho, and their inclusion relates only to the distribution records of browse occurring within these countries. A list of the publications used in compiling the database is documented in Appendix A. From this inventory, all species described as climbers, geophytes, restoids/restinoids, and succulents, as defined by Germishuizen and Meyer [25], were not considered, generating a final list of potential browsed species described as shrubs (all heights) and trees. Germishuizen and Meyer [25] were closely followed in terms of growth form, accepting the possibility that other authors describe some plants differently. Species described as seldom or minimally utilised as browse were also included in the database, e.g., some members of *Commiphora*, as well as those where the fruit is the only plant part utilised, e.g., some *Ficus* spp. A total of 722 species and infraspecific taxa, formed an integral part of the final inventory, comprising 613 species that were used to generate the distribution and descriptive data for this study.

The distribution data per quarter degree grid cell (QDGC) for each species were extracted from the South African National Biodiversity Institute's Botanical Database of South Africa (BODATSA), accessed between January and March 2022. Records from the National Herbarium in Pretoria (PRE), the Compton Herbarium in Cape Town (NBG & SAM), and the KwaZulu-Natal Herbarium in Durban (NH) are contained in BODATSA. The data was edited by removing duplicate and incomplete records. A total of 41,192 records from 1782 QDGCs were obtained and used in generating data for analysis in this study.

A Multivariate Agglomerative Hierarchical Clustering (AHC) was applied to the presence or absence of browsed species recorded in the dataset. The QDGCs with only one or two records were removed from the data set, resulting in the removal of 245 QDGCs. The statistical software PC-Ord, version 5.31 [26] was used to perform the cluster analysis, applying the Euclidean distance for dissimilarity and Ward's linkage. Previous studies with southern African grasses and legumes showed that, although PC-Ord formed disjointed groups, it was more accurate in placing QDGCs in the resulting phytochoria compared to using XLSTAT 1 Jun 2010 Software (Addinsoft to MS Excel) [24]. The geographical distribution of the clusters resulting from the AHC was compared with existing maps of well supported phytogeographical regions in the study area, especially those depicting biomes and bioregions [27] of southern Africa. This was performed to identify the phytogeographical boundaries of the clusters and to relate these to meaningful ecologically interpretable groups, subsequently referred to as browse-choria.

The QDGCs assigned to ecotones (where two or more biomes or bioregions converge in a QDGC) were not included in the dataset where the key biomes and bioregions of browse-choria are defined. Similar to previous studies [22,24], the total number of occurrences (records) in a browse-chorion was calculated, and 20 key species with the highest occurrences were selected to identify species with presumably wide ecological tolerance in a particular browse-chorion. The collection intensity (expressed as the number of browsed species per QDGC collected as herbarium specimens) was calculated and mapped on the biome and bioregion maps of South Africa, Lesotho, and Eswatini [28]. The descriptive data included growth form as described by Germishuizen and Meyer [25], duration (deciduous or evergreen), seed-bearing 'type' (fleshy fruit or dehiscent pods), utilisable plant parts, and browser type (livestock and/or game) sourced mostly from the literature recorded in Appendix A. Included in the duration dataset were the classes deciduous to evergreen,

semi-deciduous, semi-deciduous to deciduous, and semi-deciduous to evergreen. Classes with mostly single records, such as briefly deciduous, briefly deciduous to evergreen, semi-deciduous to deciduous to evergreen, and semi-evergreen, were excluded. Data for plant parts utilised are, in some cases, linked to specific utilisation studies (mostly wildlife, specifically mammals), and therefore mega-browsers such as African elephants, black rhinoceros (*Diceros bicornis*), or giraffes are the only species recorded as browsing a specific woody species. In most instances, however, plants are browsed by a combination of different browsers. Shoots were used as a collective term for young shoots, shoots, twigs, and branches. Species with different types of fruit, i.e., *Hermannia* and *Gymnosporia* with capsuled fruit and *Terminalia* and *Tetragonia* with winged fruit, were grouped as fruit-bearing. The conservational status [29] and the encroachment/invasive traits [17] of browsed species concluded the data set.

### 3. Results and Discussion

#### 3.1. Browse-Choria of South Africa, Lesotho, and Eswatini

The result of the AHC is shown as a dendrogram in Figure 1. As in previous biogeographical studies by Trytsman et al. [22,24], smaller clusters were distinguished from the dendrogram, mapped, and examined for meaningful ecologically interpretable groups. A total of 25 coherent clusters with clear phytographical boundaries were identified (Supplementary Material Figure S1) and grouped into seven distinct browse-choria as shown in Table 1. The ecologically uninformative Generalist group is henceforth excluded from further discussions. The QDGCs assigned to the Generalist group as well as those not included in the AHC (1 or 2 spp. per QDGC) were re-assigned to an appropriate cluster. The names assigned to each browse-chorion were largely derived from both the climatic regions [30,31] and biomes of South Africa, Lesotho, and Eswatini [28]. The QDGCs assigned to each cluster are presented in Figure 2, using the biome map of South Africa [28]. The outlier QDGCs were identified (circles in Figure 2) and reassigned to the most fitting browse-chorion, resulting in an all-inclusive QDGC (1782) dataset. The seven browse-choria, Central Arid (CA), Eastern Subtropical (ES), Highland Temperate (HT), Moist Temperate (MT), Northern Subtropical (NS), Southern Temperate (ST), and Western Arid (WA), will be discussed in alphabetical rather than chronological order (Table 1) since multiple clusters (except clusters 24 and 25) were merged to form a browse-chorion. In a preliminary study that included 568 species and 38,861 records, only four browse-choria were distinguished using XLSTAT 01 Jun 2010 Software (Addinsoft to MS Excel) [32]. These four browse-choria have similar distribution patterns as the ES, NS, ST, and WA browse-choria of the present study, whereas the CA, HT, and MT browse-choria were not delineated.

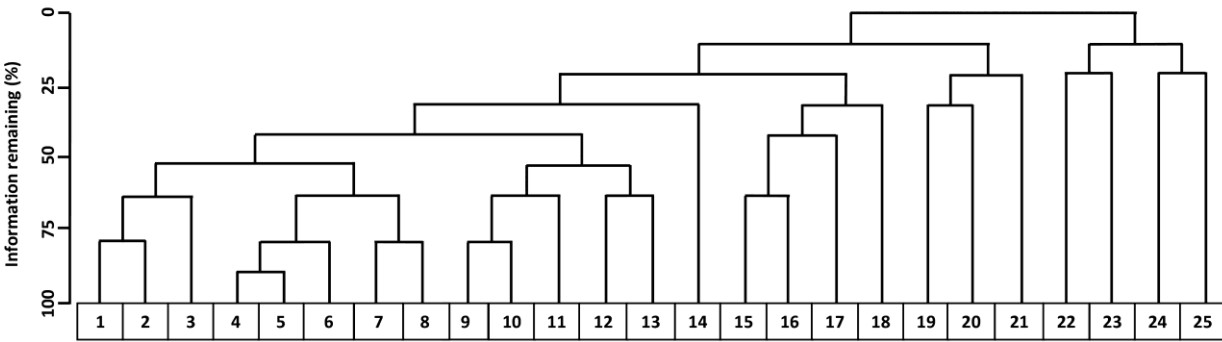

**Figure 1.** Dendrogram of browse-choria indigenous to South Africa, Lesotho, and Eswatini, delimited by Multivariate Agglomerative Hierarchical Clustering. For the grouping of the 25 numbered clusters into browse-choria, see Table 1 and Figure S1.

**Table 1.** Groups (browse-choria) formed by the clusters (see Figure 1) derived from Multivariate Agglomerative Hierarchical Clustering using PC-Ord for browse indigenous to South Africa, Lesotho, and Eswatini. For each browse-chorion, the biome(s)/region with which it is more or less congruent is supplied between brackets.

| Browse-Chorion (Biome/Region) | PC-Ord Cluster |
| --- | --- |
| Generalist | 1, 2, 4, 5, 8 |
| Western Arid (Succulent Karoo) | 3, 13, 19 |
| Southern Temperate (Fynbos, Albany Thicket) | 6, 11, 20, 21 |
| Highland Temperate (Grassland) | 7, 10, 15, 17 |
| Central Arid (Nama-Karoo) | 9, 12, 16, 18 |
| Northern Subtropical (Northern Savanna) | 14, 22, 23 |
| Moist Temperate (Eastern Great Escarpment) | 24 |
| Eastern Subtropical Coast (Indian Ocean Coastal Belt) | 25 |

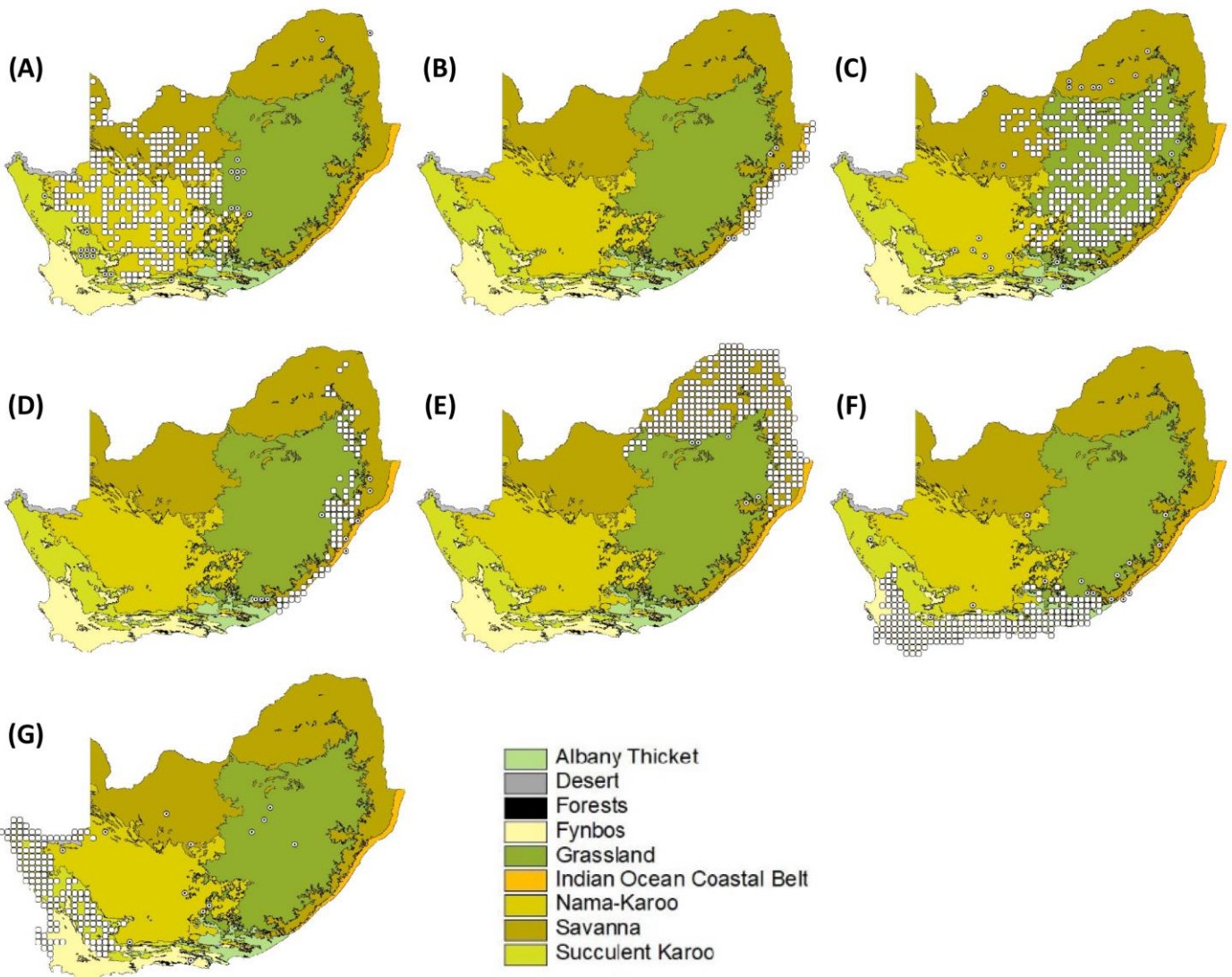

**Figure 2.** Phytochoria of browsed species in South Africa, Lesotho, and Eswatini. Core QDGSs for a particular browse-chorion are indicated with open circles. Circles with a central dot signify outlier QDGSs. Browse-choria: (**A**) Central Arid; (**B**) Eastern Subtropical; (**C**) Highland Temperate; (**D**) Moist Temperate; (**E**) Northern Subtropical; (**F**) Southern Temperate; and (**G**) Western Arid. Maps reprinted/adapted with permission from Rutherford et al. [28]. 2023, SANBI.

### 3.1.1. Central Arid Browse-Chorion (CA)

Figure 2 shows the distribution of the CA Browse-chorion, inclusive of the Karoo and *Acacia* Savanna woody vegetation types [18] and the Namib Desert and Kalahari Sand phytogeographical regions [33]. The total number of browse recorded in this browse-chorion is 306 species from 55 families. Noteworthy is the relatively low plant collection intensity, especially in the Kalahari Duneveld Bioregion. This browse-chorion includes browse mostly found in the Nama-Karoo Biome (59%), followed by the Savanna Biome (36%) (Table 2). The main bioregions include the Eastern Kalahari Bushveld (32%), and the Upper Karoo (27%), as evident from the bioregions map [28] (for the bioregions map, see Figure 3A). *Peliostomum leucorrhizum* (dwarf shrub), *Lessertia frutescens* (dwarf shrub/shrub), and *Chrysocoma ciliata* (shrub) have the highest occurrences in this browse-chorion (Table 3). Where *Peliostomum leucorrhizum* occurs mostly in the Nama-Karoo Biome, *Lessertia frutescens* and *Chrysocoma ciliata* have wide ranges, with a relatively high occurrence in the ST Browse-chorion (Figure S2). *Diospyros lycioides* is recorded as a key species present in most other browse-choria (ES, HT, MT, and NS), where *D. lycioides* subsp. *lycioides* (shrub/tree) and not *D. lycioides* subsp. *sericea* (shrub/tree) were mostly recorded (Figure S2). Key species have mostly dwarf shrub growth forms. The rationale behind the identification and formal labelling of infraspecific genetic variants (both physiological and morphological) and their importance in biogeographical studies are also highlighted here, as in Trytsman et al. [24]. Overall, the CA Browse-chorion has more key species in common with the adjacent HT Browse-chorion than with the other browse-choria. Since the number of occurrences of key species assigned to browse-choria (shown in Table 3) is not linked to biomass production or the browser's acceptability, its importance in this study is merely an indication of their wide range of tolerance to abiotic factors.

**Table 2.** Representation percentages of key biomes and bioregions (following Rutherford et al. [28]) within the browse-choria of South Africa, Lesotho, and Eswatini. Biomes and bioregions not represented were omitted. Bold-formatted values indicate the biome and bioregion with the highest percentage representation in a particular browse-chorion. Browse-choria: CA = Central Arid; ES = Eastern Subtropical; HT = Highland Temperate; MT = Moist Temperate; NS = Northern Subtropical; ST = Southern Temperate; WA = Western Arid. Biomes/Bioregions reprinted/adapted with permission from Rutherford et al. [28]. 2023, SANBI.

| Biomes/Bioregions | Browse-Chorion | | | | | | |
|---|---|---|---|---|---|---|---|
| **Biome** | **CA** | **ES** | **HT** | **MT** | **NS** | **ST** | **WA** |
| Albany Thicket | | | | 5 | | 14 | |
| Desert | | | | | | | 12 |
| Fynbos | | | | | | **84** | 15 |
| Grassland | 4 | 4 | **86** | 44 | 2 | 1 | |
| Indian Ocean Coastal Belt | | 80 | | 5 | | | |
| Nama-Karoo | **59** | | 2 | | | | 2 |
| Savanna | 36 | 16 | 12 | **46** | 98 | | |
| Succulent Karoo | 1 | | | | | 1 | **71** |
| **Bioregion** | | | | | | | |
| Albany Thicket | | | | 5 | | 23 | |
| Bushmanland | 20 | | | | | | 3 |
| Central Bushveld | | | | 1 | **49** | | |
| Drakensberg Grassland | | | 12 | | | | |
| Dry Highveld Grassland | 7 | | **25** | | | | |
| Eastern Fynbos-Renosterveld | | | | | | **27** | |
| Eastern Kalahari Bushveld | **32** | | 13 | | | | |
| Gariep Desert | | | | | | | 10 |
| Indian Ocean Coastal Belt | | **86** | | 7 | | | |
| Kalahari Duneveld | 3 | | | | | | |
| Knersvlakte | | | | | | 2 | 3 |

**Table 2.** *Cont.*

| Biomes/Bioregions | Browse-Chorion | | | | | | |
|---|---|---|---|---|---|---|---|
| **Bioregion** | **CA** | **ES** | **HT** | **MT** | **NS** | **ST** | **WA** |
| Lower Karoo | 10 | | | | | 1 | |
| Lowveld | | 6 | | 25 | 37 | | |
| Mesic Highveld Grassland | | | 24 | 15 | 3 | | |
| Mopane | | | | | 10 | | |
| Namaqualand Cape Shrubland | | | | | | | 3 |
| Namaqualand Hardeveld | | | | | | | **29** |
| Namaqualand Sandveld | | | | | | | 15 |
| Northwest Fynbos | | | | | | 16 | 1 |
| Rainshadow Valley Karoo | | | | | | 2 | 13 |
| Richtersveld | | | | | | | 7 |
| South Coast Fynbos | | | | | | 5 | |
| South Strandveld | | | | | | 2 | |
| Southern Namib Desert | | | | | | | 3 |
| Southwest Fynbos | | | | | | 18 | 2 |
| Sub-Escarpment Grassland | | | 21 | 16 | | | |
| Sub-Escarpment Savanna | | 8 | 1 | **31** | 1 | | |
| Trans-Escarpment Succulent Karoo | 1 | | | | | | 5 |
| Upper Karoo | 27 | | 4 | | | | |
| West Coast Renosterveld | | | | | | 3 | 1 |
| West Strandveld | | | | | | 1 | 5 |

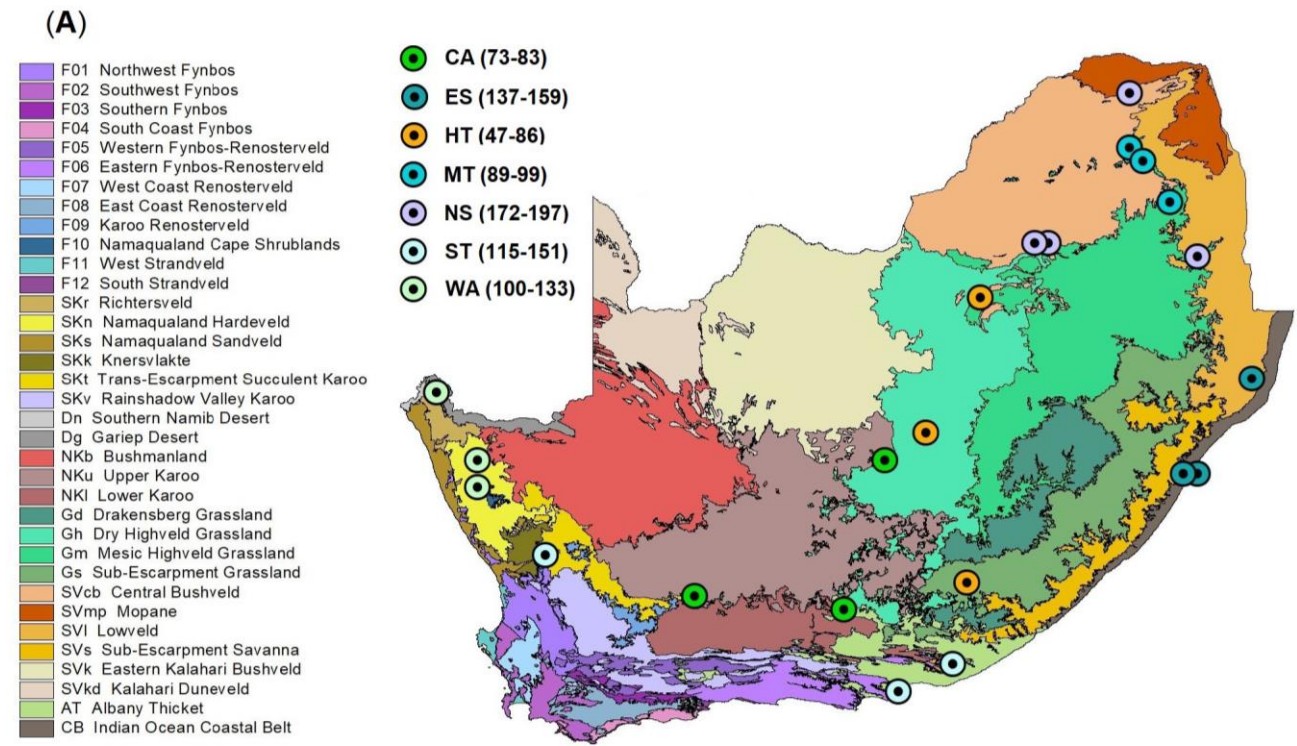

**Figure 3.** *Cont.*

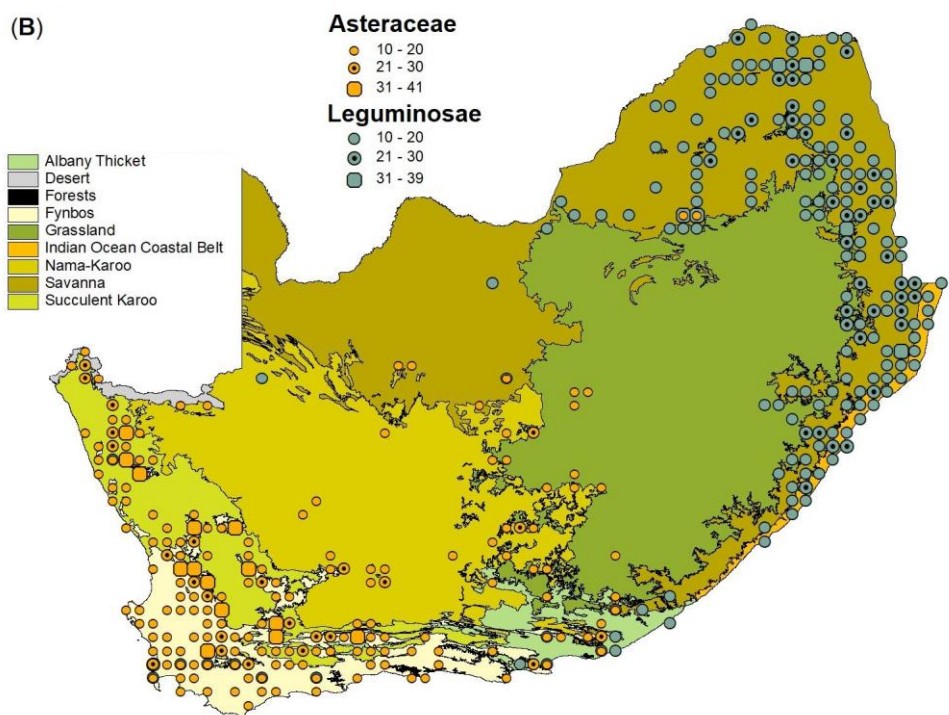

**Figure 3.** (**A**) The three highest collection intensities of browse species indigenous to South Africa, Lesotho, and Eswatini, are shown for each browse-chorion superimposed on the bioregions map [28]. Numbers in brackets represent the range (number of browse species) for the data presented. Browse-choria: CA = Central Arid; ES = Eastern Subtropical; HT = Highland Temperate; MT = Moist Temperate; NS = Northern Subtropical; ST = Southern Temperate; WA = Western Arid; (**B**) Collection intensities of Asteraceae (orange) and Leguminosae (green), depicted as the number of browse species, superimposed on the biomes map of southern Africa [28]. Maps reprinted/adapted with permission from Rutherford et al. [28]. 2023, SANBI.

**Table 3.** List of key browse species (with habit; abbreviations explained at foot of table) recorded in the browse-choria of South Africa, Lesotho, and Eswatini, with the total number of occurrences (# Occ) within each browse-chorion. Key species with superscripts are present as key species in the corresponding browse-choria (abbreviated).

| Central Arid (CA) | # Occ | Eastern Subtropical (ES) | # Occ | Highland Temperate (HT) | # Occ | Moist Temperate (MT) | # Occ |
|---|---|---|---|---|---|---|---|
| *Peliostomum leucorrhizum* (DS) | 119 | *Dichrostachys cinerea* [NS] (S/T) | 45 | *Searsia pyroides* [ES, MT, NS] (S/T) | 200 | *Searsia pyroides* [ES, HT, NS] (S/T) | 70 |
| *Lessertia frutescens* [HT, ST, WA] (DS/S) | 88 | *Mystroxylon aethiopicum* [MT, NS] (S/T) | 43 | *Diospyros lycioides* [CA, ES, MT, NS] (S/T) | 156 | *Diospyros lycioides* [CA, ES, HT, NS] (S/T) | 70 |
| *Chrysocoma ciliata* [HT, ST, WA] (S) | 81 | *Euclea natalensis* [NS] (S/T) | 41 | *Felicia muricata* [CA] (S) | 130 | *Grewia occidentalis* [ES, HT] (S/T) | 53 |
| *Lycium cinereum* (DS/S) | 78 | *Canthium inerme* (S/T) | 33 | *Diospyros austroafricana* [CA] (S) | 121 | *Lippia javanica* [NS] (S) | 53 |
| *Limeum aethiopicum* (DS/H) | 76 | *Ochna natalitia* (S/T) | 33 | *Searsia dentata* [MT] (S/T) | 116 | *Searsia dentata* [HT] (S/T) | 52 |
| *Pentzia incana* (S) | 76 | *Grewia occidentalis* [HT, MT] (S/T) | 32 | *Chrysocoma ciliata* [CA, ST, WA] (S) | 115 | *Vachellia karroo* [ES, HT, NS] (S/T) | 49 |
| *Helichrysum zeyheri* (DS/S) | 74 | *Vachellia karroo* [HT, MT, NS] (S/T) | 30 | *Gomphocarpus fruticosus* (S/H) | 106 | *Maytenus undata* (S/T) | 43 |
| *Oedera humilis* (DS) | 74 | *Searsia pyroides* [HT, MT, NS] (S/T) | 30 | *Euclea crispa* [MT] (S/T) | 98 | *Searsia rehmanniana* (S/T) | 43 |
| *Felicia muricata* [HT] (S) | 74 | *Syzygium cordatum* [MT] (S/T) | 30 | *Felicia filifolia* [ST] (S) | 89 | *Pittosporum viridiflorum* (S/T) | 41 |
| *Hermannia spinosa* (DS) | 72 | *Apodytes dimidiata* [MT] (S/T) | 30 | *Helichrysum dregeanum* (DS) | 79 | *Mystroxylon aethiopicum* [ES, NS] (S/T) | 40 |
| *Diospyros lycioides* [ES, HT, MT, NS] (S/T) | 60 | *Antidesma venosum* (S/T) | 30 | *Vachellia karroo* [ES, MT, NS] (S/T) | 71 | *Euclea crispa* [HT] (S/T) | 39 |
| *Melolobium candicans* (DS/S/H) | 56 | *Diospyros lycioides* [CA, HT, MT, NS] (S/T) | 29 | *Lessertia frutescens* [CA, ST, WA] (DS/S) | 71 | *Searsia chirindensis* (S/T) | 39 |
| *Aptosimum indivisum* (DS) | 56 | *Acalypha glabrata* [MT] (S/T) | 29 | *Pentzia globosa* (S) | 71 | *Calpurnia aurea* (S/T) | 39 |
| *Polygala leptophylla* (DS) | 55 | *Deinbollia oblongifolia* (S/T) | 29 | *Cussonia paniculata* (T) | 67 | *Senegalia ataxacantha* (S/T/C) | 39 |
| *Diospyros austroafricana* [HT] (S) | 54 | *Trema orientale* (S/T) | 29 | *Seriphium plumosum* [ST] (S) | 64 | *Senegalia caffra* (S/T) | 36 |
| *Lacomucinaea lineata* [WA] (DS/S/P) | 52 | *Sclerocroton integerrimus* (S/T) | 29 | *Gomphostigma virgatum* (DS/S/H) | 63 | *Apodytes dimidiata* [ES] (S/T) | 36 |
| *Eriocephalus ericoides* (S) | 52 | *Ekebergia capensis* (T) | 28 | *Halleria lucida* (S/T) | 60 | *Carissa bispinosa* [ES] (S) | 35 |
| *Hermannia cuneifolia* [WA] (DS) | 51 | *Bridelia micrantha* (S/T) | 28 | *Grewia occidentalis* [ES, MT] (S/T) | 59 | *Crotalaria capensis* [ES] (S/T) | 35 |
| *Pentzia lanata* (S) | 50 | *Carissa bispinosa* [MT] (S) | 27 | *Jamesbrittenia atropurpurea* [CA] (DS/S) | 57 | *Acalypha glabrata* [ES] (S/T) | 35 |
| *Jamesbrittenia atropurpurea* [HT] (DS/S) | 49 | *Crotalaria capensis* [MT] (S/T) | 26 | *Salix mucronata* (S/T) | 54 | *Diospyros whyteana* (S/T) | 34 |
| | | *Albizia adianthifolia* (T) | 26 | *Buddleja salviifolia* (S/T) | 54 | *Zanthoxylum capense* (S/T) | 34 |
| | | | | *Jamesbrittenia filicaulis* (DS) | 54 | *Plectranthus fruticosus* (S/H) | 34 |
| | | | | | | *Syzygium cordatum* [ES] (S/T) | 34 |
| **% of total number of records** | **27** | | **22** | | **42** | | **26** |

**Table 3.** *Cont.*

| Northern Subtropical (NS) | # Occ | Southern Temperate (ST) | # Occ | Western Arid (WA) | # Occ |
|---|---|---|---|---|---|
| *Dichrostachys cinerea* [ES] (S/T) | 173 | *Felicia filifolia* [HT] (S) | 138 | *Lessertia frutescens* [CA, HT, ST] (DS/S) | 83 |
| *Diospyros lycioides* [CA, ES, HT, MT] (S/T) | 169 | *Erica plukenetii* (S) | 116 | *Didelta carnosa* (DS) | 77 |
| *Croton gratissimus* (S/T) | 141 | *Eriocephalus africanus* (S) | 114 | *Pteronia divaricata* (S) | 57 |
| *Combretum apiculatum* (S/T) | 140 | *Chrysocoma ciliata* [CA, HT, WA] (S) | 106 | *Hermannia cuneifolia* [CA] (DS) | 53 |
| *Searsia pyroides* [ES, HT, MT] (S/T) | 134 | *Lessertia frutescens* [CA, HT, WA] (DS/S) | 105 | *Osteospermum sinuatum* (S) | 52 |
| *Combretum hereroense* (S/T) | 133 | *Metalasia densa* (S) | 105 | *Pteronia incana* (S) | 47 |
| *Combretum molle* (T) | 128 | *Helichrysum asperum* (DS) | 102 | *Calobota sericea* (DS/S) | 47 |
| *Ziziphus mucronata* (S/T) | 121 | *Aspalathus spinosa* (S) | 102 | *Pseudodictamnus africanus* (DS/H) | 46 |
| *Terminalia sericea* (T) | 119 | *Helichrysum rosum* (DS/S) | 91 | *Galenia fruticosa* (DS) | 46 |
| *Mundulea sericea* (S/T) | 116 | *Seriphium plumosum* [HT] (S) | 88 | *Calobota angustifolia* (DS/S) | 46 |
| *Vachellia karroo* [ES, HT, MT] (S/T) | 113 | *Searsia lucida* (S/T) | 86 | *Chrysocoma ciliata* [CA, HT, ST] (S) | 45 |
| *Euclea natalensis* [ES] (S/T) | 112 | *Indigofera heterophylla* (DS/H) | 82 | *Tetragonia fruticosa* (DS) | 45 |
| *Mystroxylon aethiopicum* [ES, MT] (S/T) | 112 | *Aspalathus hispida* (DS/S) | 80 | *Galenia sarcophylla* (DS/H) | 45 |
| *Grewia monticola* (S/T) | 109 | *Anthospermum spathulatum* (DS/S) | 79 | *Leysera gnaphalodes* (DS/S) | 42 |
| *Peltophorum africanum* (T) | 109 | *Dicerothamnus rhinocerotis* (DS/S) | 77 | *Pharnaceum aurantium* (DS) | 42 |
| *Ozoroa paniculosa* (S/T) | 107 | *Olea europaea* (S/T) | 75 | *Searsia undulata* (S) | 41 |
| *Flueggea virosa* (S/T) | 102 | *Oedera genistifolia* (S) | 75 | *Limeum africanum* (DS/H) | 41 |
| *Grewia flavescens* (S) | 102 | *Muraltia spinosa* (DS/S) | 72 | *Manochlamys albicans* (S) | 41 |
| *Lippia javanica* [MT] (S) | 102 | *Pteronia incana* (S) | 72 | *Hermannia trifurca* (DS) | 41 |
| *Ximenia caffra* (S/T) | 98 | *Dodonaea viscosa* (S/T) | 70 | *Lacomucinaea lineata* [CA] (DS/S/P) | 41 |
| | | *Maytenus oleoides* (S/T) | 70 | | |
| *% of total number of records* | *18* | | *25* | | *22* |

C = Climber, DS = Dwarf Shrub, H = Herb, P = Parasite, S = Shrub, T = Tree.

### 3.1.2. Eastern Subtropical Browse-Chorion (ES)

The ES Browse-chorion is not only relatively small compared to the other browse-choria (Figure 2) but is also the one with the lowest number of recorded browse species, namely 261 from 56 families. The Mixed Coastal Forest and Thorn Savanna were distinguished by Walker [18] as woody vegetation types, and, to some extent, the Coastal Zambesian phytogeographic region of Sayre et al. [33] represents this browse-chorion. This region is also congruent with the Northern Coastal Group [34]. The Indian Ocean Coastal Belt Biome and Bioregion represent >80% of this browse-chorion (Table 2). *Dichrostachys cinerea* (shrub/tree), *Mystroxylon aethiopicum* (shrub/tree), and *Euclea natalensis* (shrub/tree) have the highest occurrences and are also key species with high occurrences in the NS Browse-chorion (Table 3; Figure S2). The infraspecific taxa *D. cinerea* subsp. *africana* var. *africana*, *M. aethiopicum* subsp. *aethiopicum*, and *M. aethiopicum* subsp. *schlechteri* are mostly recorded here. *Diospyros lycioides* is a key species present in most other browse-choria, where *D. lycioides* subsp. *sericea* and not *D. lycioides* subsp. *lycioides* are mostly recorded (refer to distribution maps in Figure S2). The key species recorded here have mostly a shrub/tree growth form and are also present in the MT Browse-chorion.

### 3.1.3. Highland Temperate Browse-Chorion (HT)

The HT Browse-chorion has a wide distribution range (Figure 2) and is 86% enclosed by the Grassland Biome, with which it is largely congruent. It also corresponds with the Drakensberg Mountains phytogeographical region [33]. The total number of browse recorded is 328 species from 57 families. The Dry (25%) and Mesic Highveld (24%) bioregions represent this browse-chorion (Table 2). Key species, *Searsia pyroides* (shrub/tree), and *Diospyros lycioides*, have especially high occurrences (Table 3). *Searsia pyroides* var. *gracilis* has a similar distribution pattern to this browse-chorion as compared to the other two varieties, *S. pyroides* var. *integrifolia* and *S. pyroides* var. *pyroides* (Figure S2). Many of the key species recorded in the HT Browse-chorion are shrubs and are also key species in the CA and MT browse-choria.

### 3.1.4. Moist Temperate Browse-Chorion (MT)

The MT Browse-chorion is mostly found along the northeastern Great Escarpment, extending southwards into the lower sub-escarpment and coastal areas (Figure 2). The southern part of the Mixed Coastal Forest and Thorn Savanna woody vegetation types [18] as well as the Drakensberg Mountains phytogeographical region [33] represent this browse-chorion. A total of 343 browse species from 62 families are recorded here. The Savanna and Grassland biomes and the Sub-Escarpment Savanna bioregion enclose this browse-chorion (Table 2). Similar to the HT Browse-chorion, *Searsia pyroides* and *Diospyros lycioides* are the highest recorded key species, with *S. pyroides* var. *integrifolia* showing a similar distribution pattern, especially with the northern escarpment (Table 3 and Figure S2). Most of the key species have a shrub/tree growth form. The MT Browse-chorion shares many key species with the ES Browse-chorion, where five of the key species are found exclusively within these two browse-choria, for example, *Apodytes dimidiata* (Figure S2).

### 3.1.5. Northern Subtropical Browse-Chorion (NS)

Figure 2 shows the distribution of the NS Browse-chorion, enclosing most of the QDGCs within the Savanna Biome. This browse-chorion is inclusive of three broad woody vegetation types, namely, the Arid Shrub and Tree Savanna, the *Colophospermum mopane* Savanna, and the Mixed Tree and Shrub Savanna [18]. The phytogeographical region, Central African Plateau Miombo [33], is congruent with this browse-chorion. The total number of browse recorded is 387 species from 63 families, which is the highest number of species recorded in a browse-chorion. The Savanna Biome (98%) and Central Bushveld Bioregion (49%) represent the NS Browse-chorion (Table 2). In addition to high occurrences of *Dichrostachys cinerea* subsp. *africana* var. *africana* (shrub/tree) and *Diospyros lycioides*,

browse such as *Croton gratissimus* (shrub/tree) and *Combretum apiculatum* subsp. *apiculatum* are key species uniquely found in this browse-chorion (Table 3 and Figure S2). Key species listed in this browse-chorion have a shrub/tree growth form and are listed mostly as key species in the ES Browse-chorion.

3.1.6. Southern Temperate Browse-Chorion (ST)

Figure 2 shows the ST Browse-chorion that is mostly found within the Fynbos Biome (84%) (Table 2). The Cape phytogeographical region fully encloses this browse-chorion [33]. The Eastern Fynbos-Renosterveld Bioregion represents 27% and the Albany Thicket Bioregion 23% of this browse-chorion. The total number of browse recorded is 360 species from 61 families, the second-highest number of species recorded. *Felicia filifolia* (shrub), *Erica plukenetii* (shrub), and *Eriocephalus africanus* (shrub) are key species with the highest occurrences (Table 3). *Felicia filifolia* subsp. *filifolia* has a wide occurrence, even within the Mesic Highveld Grassland Bioregion, whereas *F. filifolia* subsp. *bodkinii* and *F. filifolia* subsp. *schaeferi* are confined to the Fynbos Biome (Figure S2). Of all the subspecies, *Erica plukenetii* subsp. *plukenetii* and *Eriocephalus africanus* var. *paniculatus* have the widest distribution range (Figure S2). Most of the key species have a shrub growth form, with only a few species listed as key species in the HT Browse-chorion.

3.1.7. Western Arid Browse-Chorion (WA)

The WA Browse-chorion is located mostly in the Succulent Karoo Biome (71%), but also uniquely includes the Desert Biome (Figure 2 and Table 2). The Karoo woody vegetation type [18] and the Cape and Namib Desert phytogeographical regions [33] mostly represent this browse-chorion. The Namaqualand Hardeveld Bioregion represents 29% of this browse-chorion. The number of browse recorded is 293 species from 49 families, of which *Lessertia frutescens* and *Didelta carnosa* (dwarf shrub) have the highest occurrences (Table 3). *Lessertia frutescens* subsp. *frutescens* has a higher occurrence in the WA Browse-chorion compared to *L. frutescens* subsp. *microphylla*, whereas both varieties of *Didelta* are confined to the Succulent Karoo and Fynbos biomes (Figure S2). Most of the key species have a dwarf shrub growth form. Nenzhelele et al. [35] highlight the importance of shrubs to provide a fodder bank, especially during drought in the Succulent Karoo Biome. *Calobota sericea*, with a relatively high occurrence in this browse-chorion (Figure S2), was recently investigated for its responses to abiotic stresses on germination and its nutritional quality [36,37]. Similar to the ST Browse-chorion, not many key species are present as key species in other browse-choria.

*3.2. Collection Intensity of Browsed Species Documented in South Africa, Lesotho, and Eswatini*

The three highest collection intensities for the seven browse-choria are shown in Figure 3. For the CA Browse-chorion, QDGC 3221BB has the highest number of browse, namely 83 species. This QDGC is present in the Nama-Karoo Biome or Upper Karoo Bioregion, where *Hermannia* (eight species) and *Galenia* (five species) are the most represented genera. Asteraceae is the key family with 26 species, followed by Aizoaceae with nine species. In the ES Browse-chorion, the highest number of browse is recorded in 2832AA (159 species), present in the Savanna Biome and Lowveld Bioregion. Members of *Ficus* (nine species) and *Vachellia* (eight species) are mostly present. Leguminosae (Fabaceae s.l.) represents the highest number of species (32), followed by Anacardiaceae (11). The HT Browse-chorion has high collection intensities, mostly noted in the western region, with high browse numbers in 3126DD (86 species). This QDGC is found in the Grassland Biome and Sub-Escarpment Grassland Bioregion. *Diospyros* and *Searsia* species (four species each) are mostly present. At least 13 species belonging to Asteraceae are noted for their high representation, followed by Leguminosae (seven species). The QDGC 2430AA contains the highest number of browse for the MT Browse-chorion (99 species), an ecoregion between the Grassland and Savanna biomes. *Searsia* and *Combretum* species are recorded as the most frequent, numbering five and four species, respectively. Leguminosae has the highest

number of browse, namely 15 species, followed by eight for Anacardiaceae. The NS Browse-chorion has high browse numbers in 2229DD, i.e., 197 species in the most northern location of all the QDGCs. The Savanna Biome and the ecoregion between the Central Bushveld and Mopane bioregions are found here. *Combretum* and *Ficus* have both eight representative species; however, Leguminosae is the best-represented family with 36 species. In the ST Browse-chorion, 3326BC has the highest number of browse (151 species) in a highly diverse region. Members of *Searsia* (11 species) largely outnumber the other genera, with *Euclea* and *Felicia* both with five species. Asteraceae is represented by 28 species, followed by Leguminosae with 17 species. Finally, the WA Browse-chorion is noted for high numbers of browse in the northern Succulent Karoo Biome and Namaqualand Hardeveld Bioregion, with 133 species in 2917DB. Members of *Hermannia* and *Galenia* are mostly represented, with seven and six species, respectively. Noteworthy is the large number of browse species in Asteraceae (40 species), followed by Leguminosae (12 species).

Since in the study areas Asteraceae and Leguminosae contain the majority of browse, the collection intensities for both families are shown in Figure 3. The highest intensity for Asteraceae is in the ST Browse-chorion, especially in the Northwest Fynbos Bioregion and in the WA Browse-chorion (Namaqualand Hardeveld Bioregion). Browse belonging to Leguminosae, on the other hand, is converged in the NS Browse-chorion, with a large concentration of browse in the Northern Mistbelt region (a transitional zone between the Mesic Highveld Grassland, Lowveld, and Central Bushveld bioregions) [34]. Asteraceae is an economically important family [38], with *Osteospermum* spp. including some valuable pasture species. Leguminosae, well known for its members being important pasture and browse plants, was the focus of numerous southern African studies [22,36,37,39–41]. Furthermore, Ulian et al. [42] recognised Leguminosae and Asteraceae as some of the richest families in terms of globally edible plant species used by humans, as well as members of *Ficus*, *Diospyros*, and *Grewia* as having high numbers of edible species.

### 3.3. Functional and Utilisation Attributes of Browse

The different growth forms of browsed species are summarised in Table 4. Shrubs and dwarf shrubs are the most represented growth forms in the CA and WA browse-choria, with an almost similar number of species. The key growth forms in the ES, MT, and NT browse-choria are shrub/trees and trees. The highest number of the shrub/tree growth forms is recorded in the NS Browse-chorion, namely 173 species, as well as the tree growth form, namely 111 species. Walker [18] found that the utilisation of browse decreases as the height increases, where 85% of browse was utilised from the 0–1 m layer and only 4% from the 2.5–5 m layer. In a study of African savanna woody growth forms, Zizka et al. [43] found evidence that shrubs developed a tolerance strategy to herbivory by forming a wide, multi-stemmed, and dense habit, whereas trees have adopted an escape strategy by elevating buds, leaves, and seeds above the fire-escape height.

The key plant family within each browse-chorion (Table 4) is linked to growth form, where Leguminosae (mostly shrubs/trees) is the best represented family in the ES, HT, MT, and NS browse-choria, and Asteraceae (mostly shrubs) in the CA, ST, and WA browse-choria. The NS Browse-chorion contains the highest number of species within Leguminosae (90 species) and the WA Browse-chorion within Asteraceae (101 species). In addition to the importance of Asteraceae and Leguminosae in representing large numbers of browse, Anacardiaceae with 17 *Searsia* spp. (including infraspecific taxa) and Aizoaceae with nine *Galenia* spp. contributed considerably to the high numbers of key growth forms (Table 4). Moraceae, comprising 11 *Ficus* tree species, is also worth mentioning. The other families, however, contain relatively low species numbers, and their importance as browse is less well documented.

A few browse species have diverse growth forms. For example, *Lacomucinaea lineata* (poisonous) (CA Browse-chorion) and *Thesium hystrix* (NS Browse-chorion) are described as parasitic dwarf shrubs or shrubs. These plants are xylem-feeding root parasites [44], but host ranges for either of these species have not been determined.

**Table 4.** Most represented growth forms and families within browse-choria indigenous to South Africa, Lesotho, and Eswatini. #spp. = number of species, including infraspecific taxa.

**Growth Form**

| Central Arid | #spp. | Eastern Subtropical | #spp. | Highland Temperate | #spp. | Moist Temperate | #spp. |
|---|---|---|---|---|---|---|---|
| Shrub | 90 | Shrub/tree | 130 | Shrub/tree | 124 | Shrub/tree | 150 |
| Dwarf shrub | 74 | Tree | 77 | Shrub | 63 | Tree | 89 |
| Shrub/tree | 71 | Shrub | 24 | Dwarf shrub | 48 | Shrub | 49 |
| Dwarf shrub/shrub | 45 | Dwarf shrub/shrub | 7 | Tree | 46 | Dwarf shrub/shrub | 22 |
| Dwarf shrub/herb | 18 | Shrub/climber/tree | 7 | Dwarf shrub/shrub | 34 | Dwarf shrub | 15 |
| **Northern Subtropical** | | **Southern Temperate** | | **Western Arid** | | | |
| Shrub/tree | 173 | Shrub | 108 | Shrub | 111 | | |
| Tree | 111 | Shrub/tree | 94 | Dwarf shrub | 70 | | |
| Shrub | 45 | Dwarf shrub | 73 | Dwarf shrub/shrub | 53 | | |
| Dwarf shrub/shrub | 17 | Dwarf shrub/shrub | 50 | Shrub/tree | 45 | | |
| Dwarf shrub | 12 | Tree | 28 | Dwarf shrub/herb | 15 | | |

**Family**

| Central Arid | #spp. | Eastern Subtropical | #spp. | Highland Temperate | #spp. | Moist Temperate | #spp. |
|---|---|---|---|---|---|---|---|
| Asteraceae | 88 | Leguminosae | 58 | Leguminosae | 64 | Leguminosae | 74 |
| Leguminosae | 46 | Ebenaceae | 18 | Asteraceae | 54 | Asteraceae | 33 |
| Malvaceae | 24 | Anacardiaceae | 17 | Anacardiaceae | 25 | Anacardiaceae | 27 |
| Aizoaceae | 17 | Asteraceae | 16 | Malvaceae | 21 | Ebenaceae | 18 |
| Scrophulariaceae | 16 | Rubiaceae | 14 | Ebenaceae | 16 | Celastraceae, Combretaceae | 16 |
| **Northern Subtropical** | | **Southern Temperate** | | **Western Arid** | | | |
| Leguminosae | 90 | Asteraceae | 97 | Asteraceae | 101 | | |
| Anacardiaceae | 25 | Leguminosae | 61 | Leguminosae | 45 | | |
| Asteraceae | 24 | Anacardiaceae | 24 | Malvaceae | 21 | | |
| Ebenaceae | 20 | Malvaceae | 23 | Aizoaceae | 15 | | |
| Malvaceae, Rubiaceae | 19 | Ebenaceae | 16 | Anacardiaceae, Scrophulariaceae, Amaranthaceae | 12 | | |

The majority of browsed species are perennials, with only three annual species. *Rogeria longiflora* (a shrublet and the only true annual) and *Tephrosia dregeana* (dwarf shrub/herb, annual and occasionally perennial) are mostly present in the Desert and Nama-Karoo biomes, whereas *Gomphocarpus fruticosus* subsp. *fruticosus* (shrub/herb, annual or perennial) is present in all the biomes (Figure S2). The narrow distribution patterns of *Rogeria longiflora* and *Tephrosia dregeana* in the study area (although fairly common in Namibia) could weigh against their selection as prioritised species. Hendricks et al. [45] emphasised the valuable role short-lived plants play in feed availability, however, Samuels et al. [8] cautioned on the overdependence on annual vegetation since livestock production could be at risk during drought periods.

Figure 4 illustrates the ratios between the duration (evergreen or deciduous), seed-bearing structure (fruit or pod), plant parts utilised, and browser type (livestock or game) within each browse-chorion. The dataset for browser type is most complete (84%) compared to the data set for seed-bearing structure (80%; the balance belongs mostly to shrubs within Asteraceae), followed by the duration (72%). The available data on the plant parts browsed by livestock and/or game is most limiting, i.e., recorded for only 37% of the species.

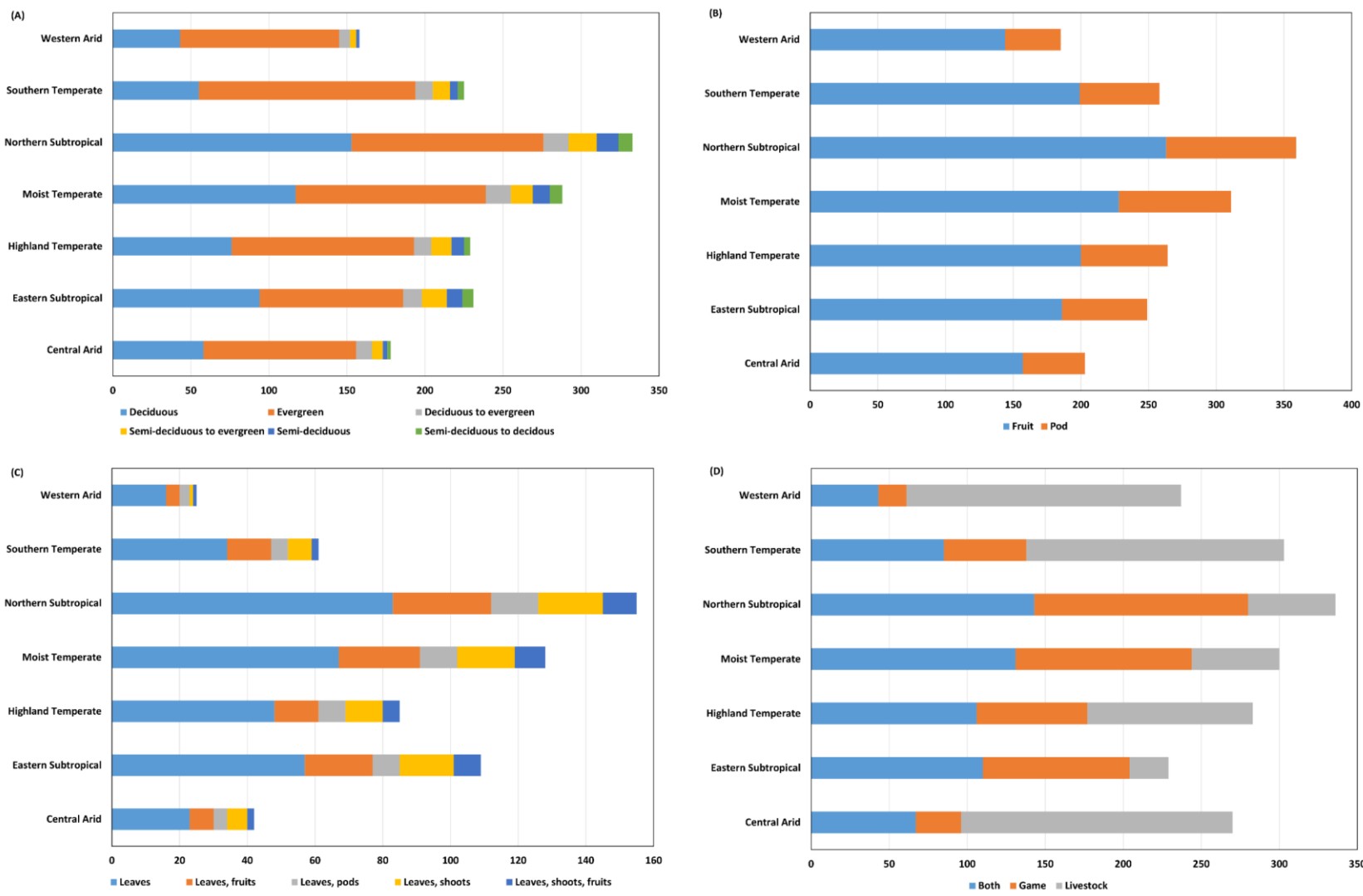

**Figure 4.** Functional and utilisation attributes of browse indigenous to South Africa, Lesotho, and Eswatini, in terms of (**A**) duration; (**B**) seed-bearing structure; (**C**) plant parts utilised; and (**D**) browser type, shown as the number of browse present in the browse-choria of southern Africa.

Evergreen browse is mostly present in all browse-choria, except in the NS Browse-chorion, where deciduous species are mostly present (Figure 4A). Zizka et al. [43] affirm that due to a higher cavitation risk (in the xylem) in seasonally dry environments, taller plants are more often deciduous and use water more efficiently. The NS Browse-chorion, furthermore, shows the highest diversity in duration, i.e., semi-deciduous, semi-deciduous to deciduous, and semi-deciduous to evergreen. Penderis [46] established that evergreen tree species in the semi-arid southern African Savanna (e.g., NS Browse-chorion) produce higher annual leaf and green shoot material compared to deciduous trees, but that the seasonal crude protein content of semi-deciduous and deciduous species is higher than in evergreen species. Watson and Owen-Smith [12] confirmed that common eland (*Taurotragus oryx*) in the Mountain Zebra National Park in South Africa preferred evergreen dwarf shrubs for at least a month during the dry winter season compared to the senescent leaves available on deciduous species. In the Grassland Biome, Janecke and Smit [47] found that semi-deciduous shrubs retained their leaves for longer compared to trees and were therefore important food sources for browsers during the critical dry winter period. Dziba et al. [48], however, concluded that in the Eastern Cape, shoot morphology and not duration determine intake by goats. As pointed out by Basha [49], the seasonal variation in the availability of browse often confounds studies on the diet selection of browsers.

The majority of browse is fruit-bearing (females only in the case of dioecious species), represented mostly by *Searsia* (dioecious) and *Ficus* spp., with a high occurrence in the NS Browse-chorion (Figure 4B). The genera *Vachellia*, *Senegalia*, *Indigofera*, and *Combretum* contributed mostly to pod-bearing browse. Bunney [50] studied tree seed dispersal spectra of megafaunal fruit across South African biomes and concluded that fleshy fruit occurs mostly in the north-eastern and eastern coastline regions (NS and ES browse-choria), while dry fruit (mostly leguminous pods) occurs in the northern central regions (CA Browse-chorion). The importance of vertebrates, especially the African elephant, as seed dispersal agents for trees across all biomes is further highlighted in this study. Pods and other fruit types are also confirmed as valuable food resources for browsers, where all sizes of dried pods and fruit provide prolonged food sources [18]. An added advantage to pods being valuable food sources and available over a longer period compared to fleshy-fruited browse is the option to collect and store pods for use during drought periods [51].

In terms of plant parts utilised by browsers, leaves are mostly consumed, followed by leaves and fruit, then leaves and shoots (Figure 4C). This trend is noted for all browse-choria. *Adansonia digitata* is recorded as the only species to be fully utilised, i.e., leaves, shoots, flowers, fruit, and seed. The authors of [42] also list this species as a neglected and underutilised plant. According to Walker [18], leaves and young shoots provide the bulk of browse; however, the proportion of these plant parts compared to the total woody biomass is very low, i.e., <1%. Leaf size also plays a role in plant part selection in semi-arid to arid savannas, where broad-leaved browse is selected during the wet seasons; however, since fine-leaved browse retains its leaves, it is mostly utilised during the dry seasons [52]. It should be mentioned that 87% of the data presented on the plant parts utilised are from only two growth forms, namely shrubs-trees and trees, with little or no records for dwarf shrubs or shrublets, an obvious deficiency of the available data. Walker [18] suggested that browsers utilise considerably more of the total annual growth from these smaller and less woody growth forms.

Game is recorded to utilise browse predominantly more as compared to livestock in the NS Browse-chorion, a pattern also noted for the MT and ES browse-choria (Figure 4D). The browser types most noted for utilisation in the CA, ST, and WA browse-choria, however, are largely livestock. The plant genera with members mostly recorded to be browsed by livestock are *Hermannia* (15 species), *Galenia* (10 species), and *Pteronia* (nine species), by game *Searsia* (seven species), *Vachellia*, *Senegalia*, and *Albizia* (six species in total), and by both types *Searsia* and *Vachellia* (both nine species), and *Combretum*, *Ficus*, and *Grewia* (seven species in total). The authors of [53] observed gender differences in the diet selection of

kudu, highlighting the complex interaction between browse and browsers and henceforth the management thereof.

### 3.4. Endangered and Invasive Attributes of Browsed Species

A list of endangered browse is presented in Table 5. *Warburgia salutaris*, a perennial shrub/tree found mostly in the NS Browse-chorion (Figure S2), is overexploited by humans for the medicinal value of, especially, its bark [54]. *Rhynchosia emarginata*, a perennial shrub, has a narrow distribution in the northern region of the WA Browse-chorion (Figure S2), whereas *Eriocephalus microphyllus* var. *carnosus* has only one botanical record in the WA Browse-chorion. Most of the near-threatened species are recorded in the ST and WA browse-choria, except for *Elaeodendron transvaalense*. The vulnerable species are found mostly in the CA and ST browse-choria with only *Searsia batophylla* recorded in the NS Browse-chorion. With the exception of the species listed in Table 5, all other species included in our analyses are classified as "Least concern" [29], and browsing is thus unlikely to have a detrimental effect on the conservation of the vast majority of species. Furthermore, the traditional medicinal and related biocultural usage of these endangered species by humans probably poses a greater threat to their survival than herbivorous animals [54–56]. A plant conservation strategy, published in 2015 by the South African National Biodiversity Institute and the Botanical Society of South Africa [57], aims to continuously conserve threatened plants both in situ and ex situ. This strategy incorporates priority sites into protected areas and promotes seed collections in collaboration with the Millennium Seed Bank Partnership.

**Table 5.** Red data list of endangered and vulnerable browse taxa [29] indigenous to South Africa, Lesotho, and Eswatini, with their presence noted in key biomes and browse-choria.

| Browsed spp. | Red List Status | Key Biome | Browse-Chorion * |
|---|---|---|---|
| *Warburgia salutaris* | Endangered | Savanna | ES, MT, NS |
| *Rhynchosia emarginata* | Endangered | Succulent Karoo | WA |
| *Eriocephalus microphyllus* var. *carnosus* | Endangered | Ecoregion | WA |
| *Elaeodendron transvaalense* | Near threatened | Savanna | NS, MT, HT, ES |
| *Passerina ericoides* | Near threatened | Fynbos | ST, WA |
| *Helichrysum tricostatum* | Near threatened | Fynbos | ST, WA |
| *Vexatorella alpina* | Near threatened | Fynbos | ST, WA |
| *Aspalathus angustifolia* subsp. *robusta* | Vulnerable | Fynbos | CA, ST, WA |
| *Searsia batophylla* | Vulnerable | Savanna | MT, NS |
| *Otholobium rotundifolium* | Vulnerable | Fynbos | ST |
| *Justicia orchioides* subsp. *orchioides* | Vulnerable | Albany Thicket | CA, ST |
| *Galenia crystallina* var. *maritima* | Vulnerable | Succulent Karoo | CA, WA |

* Browse-choria: CA = Central Arid; ES = Eastern Subtropical; HT = Highland Temperate; MT = Moist Temperate; NS = Northern Subtropical; ST = Southern Temperate; WA = Western Arid.

The collection intensity of browse, classified as woody encroachers/invaders, is mapped in Figure 5, with the highest number of such species found in the NS Browse-chorion (>19 species per QDGC). The lower numbers (less than six species per QDGC) are recorded mostly in the ST and HT browse-choria, with the Nama-Karoo (inclusive of the CA Browse-chorion) least affected. A list of these declared woody encroachers is presented in Table S1. The majority of invasive browse found in the NS Browse-chorion hotspot are mostly species of *Senegalia* and *Vachellia*. *Senegalia mellifera*, mostly present in the CA and NS browse-choria (Figure S2), is considered one of the most aggressive encroaching species, especially in the drier areas of the savanna [15,58]. *Vachellia karroo*, on the other hand, is the key encroacher species in grassland [59]. The ST Browse-chorion has *Azima tetracantha*, *Dodonaea viscosa* var. *angustifolia*, and *Seriphium plumosum* recorded as the main encroacher species. Overgrazing and the resultant colder fires and increase in seed dispersal of woody plants, fire suppression, directional changes in temperature and precipitation towards

increased aridity, as well as an increase in water availability for deep-rooted woody plants, have been suggested as some of the main drivers for encroaching woody plants [15,60].

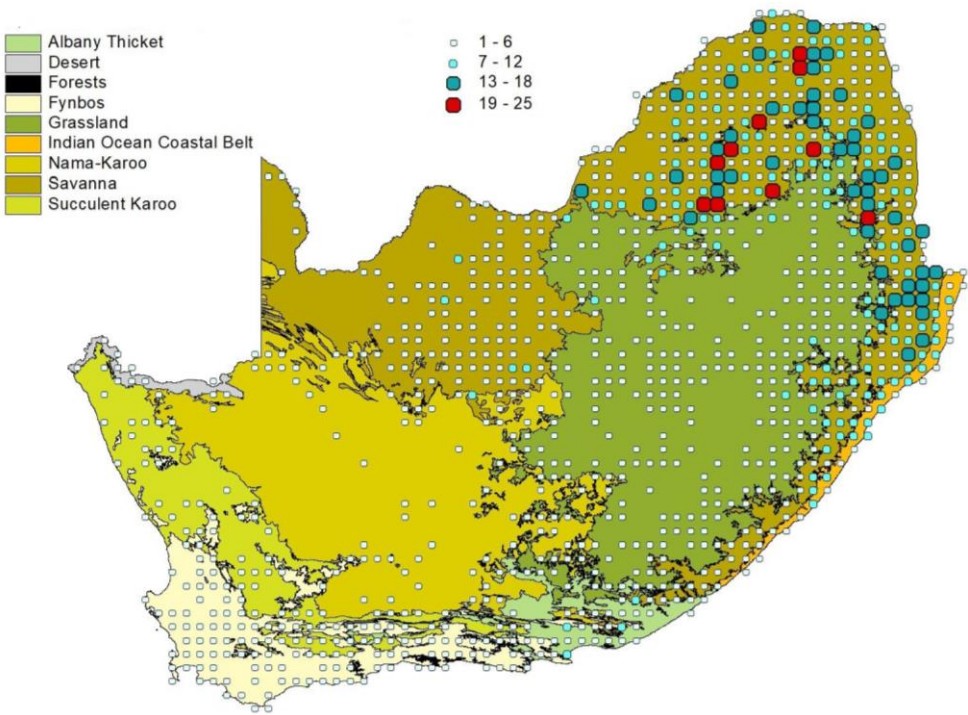

**Figure 5.** The collection intensities of browse species indigenous to South Africa, Lesotho, and Eswatini, classified as potential woody encroachers. Map reprinted/adapted with permission from Rutherford et al. [28]. 2023, SANBI.

## 4. Conclusions

It is evident from this study that, within each browse-chorion, a considerable wealth of indigenous woody species, utilised by both livestock and game, exists, and the need to further assess these valuable browse genetic resources is evident. The Eastern Kalahari Bushveld, Indian Ocean Coastal Belt, Dry Highveld Grassland, Sub-Escarpment Savanna, Central Bushveld, and Namaqualand Hardeveld were identified as highly diverse bioregions for the Central Arid, Eastern Subtropical, Highveld Temperate, Moist Temperate, Northern Subtropical, and Western Arid browse-choria, respectively. The continuance of updating the database as new browse and/or functional and utilisation attributes become available is essential, as is adding data on preference, nutritive value, and the presence of secondary compounds. Key species identified in the present study, as well as high-valued browse documented in the literature, can thus be linked for each browse-chorion with other preferred attributes. Since endangered and threatened browse is mostly present in the Northern Subtropical, Southern Temperate, and Western Arid browse-choria, the need to collect and conserve seed from these choria should be prioritised. Furthermore, additional studies are needed to evaluate the impacts of climate change on the ecological tolerance of the browse species. Such evidence will assist in predicting the possible spread of the species into new regions as well as identifying populations at risk, from which collections should be prioritised for conservation purposes. The South African National Forage Genebank could therefore become a principal source for indigenous browse genetic resources in southern Africa. On account of the great diversity of browse available to pastoralists in southern Africa, its huge value as a resource, not only from a nutritional perspective but also in terms of animal health, becomes clear. Where livestock and game have access to a diversity of species, it, in general, means a better-balanced and more sustainable diet for all.

Future studies should assess the browse-choria at local landscape scales, determine their immediate threats, and work towards finding possible solutions. There is also an

opportunity to document how pastoralists protect and conserve browse and merge these practices with indigenous knowledge as key factors for ensuring a sustainable future for the indigenous browse of South Africa, Lesotho, and Eswatini.

**Supplementary Materials:** The following supporting information can be downloaded at: https://www.mdpi.com/article/10.3390/d15070876/s1, Figure S1: Clusters formed by AHC assigned to a generalist group and seven browse-choria with clear geographical boundaries; Figure S2: Distribution patterns of key, annual, endangered, and invasive browse of South Africa, Eswatini, and Lesotho; Table S1. A list of declared woody encroacher species, indigenous to South Africa, Eswatini, and Lesotho, browsed by livestock and game.

**Author Contributions:** Conceptualization, M.T., A.E.v.W. and F.L.M.; methodology, M.T. and A.E.v.W.; validation, M.T., F.L.M., A.E.v.W., M.I.S. and C.F.C.; formal analysis, M.T.; investigation, M.T. and A.E.v.W.; resources, M.T., A.E.v.W., C.F.C. and M.I.S.; data curation, M.T. and F.L.M.; writing—original draft preparation, M.T.; writing—review and editing, M.T., A.E.v.W., F.L.M., M.I.S. and C.F.C.; project administration, F.L.M.; funding acquisition, F.L.M. All authors have read and agreed to the published version of the manuscript.

**Funding:** This research was funded by the Red Meat Research and Development (RMRD) fund of South Africa, grant number P02000229.

**Data Availability Statement:** All data presented in this manuscript are available from the corresponding author upon reasonable request.

**Acknowledgments:** The authors would like to thank Beppi Hart from the ARC for her assistance with obtaining relevant literature used in this study and Elsa van Niekerk for the graphics.

**Conflicts of Interest:** The authors declare no conflict of interest. The funders had no role in the design of the study; in the collection, analyses, or interpretation of data; in the writing of the manuscript; or in the decision to publish the results.

## Appendix A

Literature accessed to compile an inventory of browse indigenous to South Africa, Lesotho, and Eswatini.

Basha, N.A.D. Feeding Behaviour, Diet Selection of Goats and Nutritive Value of Browse Species in Sub-Humid Subtropical Savannah, South Africa. Ph.D. Thesis, University of KwaZulu-Natal, South Africa, 2012.

Brown, D.H. The Feeding Ecology of the Black Rhinoceros (*Diceros bicornis minor*) in the Great Fish River Reserve, Eastern Cape Province, South Africa. Master's Thesis, University of Fort Hare, Alice, South Africa, 2008.

Chepape, R.M.; Mbatha, K.R.; Luseba, D. Local use and knowledge validation of fodder trees and shrubs browsed by livestock in Bushbuckridge area, South Africa. *Livest. Res. Rural. Dev.* **2014**, *77*, 20–47.

Ganqa, N.M.; Scogings, P. Forage quality, twig diameter, and growth habit of woody plants browsed by black rhinoceros in semi-arid subtropical thicket, South Africa. *J. Arid Environ.* **2007**, *70*, 514–526. https://doi.org/10.1016/j.jaridenv.2007.02.003.

Gerber, J. Impacts of impala on subtropical thicket in the Shamwari Game Reserve, Eastern Cape. Master's Thesis, Nelson Mandela Metropolitan University, Gqeberha, South Africa, 2006.

Hall-Martin, A.J.; Erasmus, T.; Botha, B.P. Seasonal variation of diet and faeces composition of black rhinoceros *Diceros bicornis* in the Addo Elephant National Park. *Koedoe* **1982**, *25*, 63–82. https://doi.org/10.4102/koedoe.v25i1.605.

Haschick, S.L.; Kerley, G.I.H. Browse intake rates by bushbuck (*Tragelaphus scriptus*) and boergoats (*Capra hircus*). *Afr. J. Ecol.* **1997**, *35*, 146–155.

Hassen, A.; Rethman, N.F.G.; Apostolides, Z.; Van Niekerk, W.A. Forage production and potential nutritive value of 24 shrubby *Indigofera* accessions under field conditions in South Africa. *Trop. Grassl.* **2008**, *42*, 96–103.

Hendricks, H.H.; Novellie, P.A.; Bond, W.J.; Midgley, J.J. Diet selection of goats in the communally grazed Richtersveld National Park. *Afr. J. Range For. Sci.* **2002**, *19*, 1–11. https://doi.10.2989/10220110209485769.

Hooimeijer, J.F.; Jansen, F.A.; De Boer, W.F.; Wessels, D.; Van der Waal, C.; De Jong, C.B.; Otto, N.D.; Knoop, L. The diet of kudus in a mopane dominated area, South Africa. *Koedoe* **2005**, *48*, 93–102. https://doi.org/10.4102/koedoe.v48i2.96.

Samuels, M.I.; Cupido, C.F.; Swarts, M.B.; Palmer, A.R; Paulse, J.W. Feeding ecology of four livestock species under different management in a semi-arid pastoral system in South Africa. *Afr. J. Range For. Sci.* **2016**, *33*, 1–9. https://doi.org/10.2989/10220119.2015.1029972.

Kotze, D.C.; Zacharias, P.J.K. Utilization of woody browse and habitat by the black rhino (*Diceros bicornis*) in western Itala Game Reserve. *Afr. J. Range For. Sci.* **1993**, *10*, 36–40. https://doi.10.1080/10220119.1993.9638319.

Kunene, N.; Wilson, R.A.C.; Myeni, N.P. The use of trees, shrubs and herbs in livestock production by communal farmers in northern KwaZulu-Natal, South Africa. *Afr. J. Range For. Sci.* **2003**, *20*, 271–274.

Le Roux, P.M.; Kotze, C.D.; Glen, H.F. *Bossieveld: Grazing Plants of the Karoo and Karoo-like Areas*; Bulletin of the Department of Agriculture: Pretoria, South Africa, 1994; Volume 428.

Louw, G.; Beukes, T. Ken ons veldplante. *Die Noordwester* **1988**, 6–12.

Loveridge, J.P.; Moe, S.R. Termitaria as browsing hotspots for African megaherbivores in Miombo Woodland. *J. Trop. Ecol.* **2004**, *20*, 337–343.

Lukhele, M.S. The chemical composition and nutritive value of leaves of indigenous fodder trees. Ph.D. Thesis, University of Pretoria, Pretoria, South Africa, 2002.

Maroyi, A. Diversity of use and local knowledge of wild and cultivated plants in the Eastern Cape province, South Africa. *J. Ethnobiol. Ethnomed.* **2017**, *13*, 43. http://doi.10.1186/s13002-017-0173-8.

Mbatha, K.R.; Bakare, A.G. Browse silage as potential feed for captive wild ungulates in southern Africa: A review. *Anim. Nutr.* **2018**, *4*, 1–10.

Mthi, S.; Rust, J.M.; Morgenthal, T.L. Partial nutritional evaluation of some browser plant species utilized by communal livestock in the Eastern Cape Province, South Africa. *Appl. Anim. Husb. Rural Dev.* **2016**, *9*, 25–30.

Mudau, H.S.; Mokoboki, H.K.; Ravhuhali, K.E.; Mkhize, Z. Nutrients profile of 52 browse species found in semi-arid areas of South Africa for livestock production: Effect of harvesting site. *Plants* **2021**, *10*, 2127. https://doi.10.3390/plants10102127.

Nkqubela, M.G.; Scogings, P.F.; Raats, J.G. Diet selection and forage quality factors affecting woody plant selection by black rhinoceros in the Great Fish River Reserve, South Africa. *S. Afr. J. Wildl. Res.* **2005**, *35*, 77–83.

Nortje, J.M.; Van Wyk, B-E. Useful plants of Namaqualand, South Africa: A checklist and analysis. *S. Afr. J. Bot.* **2019**, *122*, 120–135.

Orwa, C.; Mutua, A.; Kindt, R.; Jamnadass, R.; Simons, A.J. *Agroforestree Database: A Tree Reference and Selection Guide Version 4.0*; World Agroforestry Centre, Nairobi, Kenya, 2009.

Owen-Smith, N.; Cooper, S.M. Palatability of woody plants to browsing ruminants in a South African savanna. *Ecology* **1987**, *68*, 319–331.

Palmer, E.; Pitman, N. *Trees of South Africa*; AA Balkema: Cape Town, South Africa, 1961.

Penderis, C.A. Browse: Quantity and Nutritive Value of Evergreen and Deciduous Tree Species in Semi-Arid Southern African Savannas. Ph.D. Thesis, University of KwaZulu-Natal, Durban, South Africa, 2012.

Ravhuhali, K.E.; Mlambo, V.; Beyene, T.S.; Palamuleni, L.G. Effects of soil type on density of trees and nutritive value of tree leaves in selected communal areas of South Africa. *S. Afr. J. Anim. Sci.* **2020**, *50*, 88–98. http://dx.doi.org/10.4314/sajas.v50i1.10.

South African National Biodiversity Institute. *Plantzafrica*. Available online: http://pza.sanbi.org/.

Teague, W.R.; Trollope, W.S.W.; Aucamp, A.J. Veld management in the semi-arid bush-grass communities of the Eastern Cape. *Proc. Ann. Congr. Grassl. Soc. S. Afr.* **1981**, *16*, 23–28. https://doi.10.1080/00725560.1981.9648915.

Theart, J.J.F.; Hassen, A.; Van Niekerk, W.A.; Gemeda, B.S. In-vitro screening of Kalahari browse species for rumen methane mitigation. *Sci. Agric.* **2015**, *72*, 478–483.

Van Wyk, A.E.; Van Wyk, P. *Field Guide to Trees of Southern Africa*; Penguin Random House: Cape Town, South Africa, 2013.

Van Wyk, P. *Field Guide to the Trees of the Kruger National Park*; Struik: Cape Town, South Africa, 2008.

Venter, F.; Venter, J-A. *Making the Most of Indigenous Trees*; Briza: Pretoria, South Africa, 2015.

Walker, B.H. A review of browse and its role in livestock production in southern Africa. In *Browse in Africa: The Current State of Knowledge*; Le Houérou, H.N., Ed.; International Livestock Center for Africa: Addis Ababa, Ethiopia, 1980; pp. 223–231.

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
