# Peer review of "A Phytogeographical Classification and Survey of the Indigenous Browse Flora of South Africa, Lesotho, and Eswatini"

_diversity, doi:10.3390/d15070876_

Round 1

Reviewer 1 Report

The manuscript under review presents data on the regional diversity and distribution of valuable fodder trees and shrubs from the rangelands in South Africa, Lesotho, and Eswatini. Numerous literature sources and databased were accessed to compile a database of plant species that were recorded here and utilized by ruminants and non-ruminants. Distribution data of 613 plant species from 76 families were analyzed with numerical techniques and seven distinct phytochoria were established. Authors characterized each chorion with respect to its key families and species, their functional and utilization as well as endangered and invasive attributes of the browsed species.

In my opinion, that is a straightforward study that is well planned, performed and presented. Authors used an adequate analytical methods and comprehensive data set. In my opinion this contribution could be interesting to a broad range of specialists. It broadens existing knowledge on the plant diversity, phytogeography, and conservation of South Africa.

Author Response

The manuscript was accepted, and no revisions were made.

Reviewer 2 Report

Greater emphasis should be placed on the conservation status of species

Analysis of the impact of use on the conservation status of the taxon is lacking

proofreading of the scientific language

Author Response

A paragraph was added under 3.4. Endangered and invasive attributes of browsed species, adding three new references.

Reviewer 3 Report

Overall, the work is interesting, however, the introduction should be concise and better to focus on the research topic from the beginning. The introduction speaked much useless content. 

The references format in many places is wrong. You can't say e.g. "the work has been conducted by [3,5]. I have listed some of these places, e.g. 1. L98, L118, L132, L169, L188, L197, L236, L219, L220, L407, L442, L450.

Author Response

The introduction was shortened, removing one sentence in paragraph 1 and the whole of paragraph 3 (a total of 242 words).

Author Response

As suggested, the format of one reference was changed.  (Unpublished report as referred to by Turpie et al. [31] changed to [17]

Round 2

Reviewer 3 Report

The author has modified the relevant content as requested, and I agree to accept this article for publication.